# High Pinning Force Values of a Fe(Se, Te) Single Crystal Presenting a Second Magnetization Peak Phenomenon

**DOI:** 10.3390/ma14185214

**Published:** 2021-09-10

**Authors:** Armando Galluzzi, Krastyo Buchkov, Vihren Tomov, Elena Nazarova, Antonio Leo, Gaia Grimaldi, Massimiliano Polichetti

**Affiliations:** 1Department of Physics “E.R. Caianiello”, University of Salerno, Via Giovanni Paolo II, 132, 84084 Fisciano (SA), Italy; aleo@unisa.it; 2CNR-SPIN Salerno, Via Giovanni Paolo II, 132, 84084 Fisciano (SA), Italy; gaia.grimaldi@spin.cnr.it; 3Institute of Solid State Physics, Bulgarian Academy of Sciences, 72 Tzarigradsko Chaussee Blvd., 1784 Sofia, Bulgaria; buchkov@issp.bas.bg (K.B.); vixren@gmail.com (V.T.); nazarova@issp.bas.bg (E.N.); 4Institute of Optical Materials and Technologies, Bulgarian Academy of Sciences, Acad. G. Bonchev Str. Bl. 109, 1113 Sofia, Bulgaria

**Keywords:** iron-based superconductors, dc magnetic properties, second magnetization peak phenomenon, pinning force, magnetism and superconductivity

## Abstract

The magnetization M of an Fe(Se, Te) single crystal has been measured as a function of temperature T and dc magnetic field H. The sample properties have been analyzed in the case of a magnetic field parallel to its largest face H||ab. From the M(T) measurement, the T_c_ of the sample and a magnetic background have been revealed. The superconducting hysteresis loops M(H) were between 2.5 K and 15 K showing a tilt due to the presence of a magnetic signal measured at T > T_c_. From the M(H) curves, the critical current density J_c_(H) has been extracted at different temperatures showing the presence of a second magnetization peak phenomenon. By extracting and fitting the J_c_(T) curves at different fields, a pinning regime crossover has been identified and shown to be responsible for the origin of the second magnetization peak phenomenon. Then, the different kinds of pinning centers of the sample were investigated by means of Dew-Hughes analysis, showing that the pinning mechanism in the sample can be described in the framework of the collective pinning theory. Finally, the values of the pinning force density have been calculated at different temperatures and compared with the literature in order to understand if the sample is promising for high-current and high-power applications.

## 1. Introduction

In 2008, the discovery of the iron-based superconductors (IBSs) [1] was received with great interest by the scientific community primarily because it was largely believed that magnetism and superconductivity could not coexist. After the first studies on these new compounds, it was clear that they seemed to overcome the HTS weak points. In fact, the IBSs showed low anisotropy values [2,3,4,5,6,7] and a preferable superconductor–normal–superconductor (SNS) behavior of the grain boundary junctions [8,9,10]. Despite their low T_c_ values, it has been demonstrated that the IBSs can be suitable for magnet and wire production and/or high-power applications and high-current transport thanks to their high values of critical current density J_c_, irreversibility field and upper critical field [11,12,13,14,15,16] as well as their good inter-grain connectivity [8,13,17,18]. Among the various IBS families, the 11 family has attracted a lot of interest due to its very simple crystalline structure and to the possibility of easily doping it with several elements of the periodic table [19,20,21,22,23] in order to improve the superconducting properties of the compounds. Among the compounds of the 11 family, Fe(Se, Te) is one of the most studied compounds in recent years due to its relatively high T_c_ (for single crystals between 12 K and 14.5 K), its chemical stability and also because it does not present poisonous elements in its stoichiometry. Moreover, the high values of critical current density and upper critical field makes this compound appealing in view of power applications [11,24,25,26]. Among the features of Fe(Se, Te), there is a rich variety of vortex phenomena together with the presence of particular pinning structures (such as columnar defects, twin boundaries, etc.) which can generate the interesting second magnetization peak phenomenon. It is characterized by an anomalous increasing trend of J_c_ with increasing magnetic fields [27,28,29,30,31,32,33] which attracts even more interest to the samples presenting this phenomenon due to their capability to sustain high currents at high magnetic fields. In this work, we present an analysis of the pinning properties of a single crystal having twin boundaries in the case of a magnetic field applied along its largest face (H||ab). Based on our previous studies, the second magnetization peak phenomenon manifests when the field is applied perpendicular to its largest face (H||c) [34], and we explore further the vortex behavior of the material in the H||ab field configuration in terms of this phenomenon and its associated pinning features. First, we obtained the T_c_ of the sample by means of an M(T) measurement. Then, we extracted the critical current densities as a function of field from the superconducting hysteresis loops at different temperatures. After that, by fitting the established functional dependencies of J_c_(T) at different magnetic fields within the frames of the Kim model and the Dew-Hughes pinning force scaling approach, the different kinds of pinning centers of the Fe(Se, Te) single crystal have been analyzed. In addition, we have identified surface (planar) type pinning centers in certain field and temperature ranges. Finally, starting from the J_c_(H) curves, the pinning force density values at different temperatures were calculated and compared with values reported in the literature, confirming the suitability of this material for high-power applications.

## 2. Materials and Methods

An FeSe_0.5_Te_0.5_ twinned single crystal sample with dimensions 3 × 3 × 0.2 mm^3^ (*a* × *b* × *c*) fabricated by means of the Bridgman technique was analyzed. The fabrication details are reported elsewhere [34]. By means of SEM-EDX analysis, a slightly deviated final stoichiometry Fe_0.96_Te_0.59_Se_0.45_ was found and it is typical for the crystal growth and synthesis in FeSeTe [35,36,37,38] and in the basic compound FeSe [39,40,41]. The sample was characterized using dc magnetic measurements in “parallel field configuration”, i.e., with the magnetic field applied parallel to its largest face (H||ab). The magnetization as a function of the temperature M(T) and of the magnetic field M(H) was measured by means of a Quantum Design PPMS-9T equipped with a VSM option. The residual trapped field inside the PPMS dc magnet was reduced below 1 × 10^−4^ T before each measurement following the procedure reported elsewhere [42,43]. The M(T) measurement was performed in zero field cooling (ZFC)-field cooling (FC) conditions. In particular, the sample was cooled down to 2.5 K in a zero magnetic field, then a field of 0.01 T was switched on and the data were acquired for increasing temperatures up to 300 K. After that, the sample was cooled down while acquiring FC magnetization. In terms of the M(H) measurements, the sample was cooled down to the measurement temperature in the absence of field and thermally stabilized for about 30 min. Then, the magnetic field was ramped with a sweep rate equal to 0.01 T/s to reach +9 T, then back to −9 T, and finally to +9 T again to acquire the complete hysteresis loop.

The pinning force F_p_ values, expressed in N/m^3^, were calculated at different temperatures using the formula $F_{P}=J_{c}B$ where B is the applied magnetic field H expressed in T.

## 3. Results and Discussion

The superconducting critical temperature T_c_ of the sample was obtained by performing a M(T) measurement in zero field cooling (ZFC)-field cooling (FC) conditions with an applied field of 0.01 T. The result is shown in Figure 1. The T_c_ was determined as the value of the temperature corresponding to the onset of the ZFC branch. As indicated by a red arrow in the inset of Figure 1, where an enlargement of the curve in the region of the superconducting transition is reported, this value is approximately 14.5 K, in agreement with the literature [38,44,45,46,47]. It is worth underlining the presence of a non-zero signal above T_c_ in the ZFC magnetization together with a magnetic irreversibility between the ZFC and FC curves (indicated by a double arrow in the inset of Figure 1) usually associated with a magnetic background. This could be due to magnetic impurities present in the sample as already reported for Fe(Se, Te) [35,36,37,38]. It is worth underlining that the magnetic background width in the H||ab configuration (this article) is about 15 times larger than the H||c configuration reported in Ref. [34] on the same sample. This could be ascribed to the fact that the material is magnetically anisotropic and that the magnetic signal is more activated when the field is parallel to the ab face.

To investigate the superconducting and pinning properties of the sample, the M(H) measurements were performed at different temperatures in the range between 2.5 K and 15 K. In the main panel of Figure 2, the superconducting hysteresis loops are reported. It is important to underline that the curves are slightly tilted due to the presence of the magnetic background. To explore the contribution of the magnetic background to the overall signal, the M(H) curve just above T_c_, i.e., T = 15 K, was measured and is shown in the inset of Figure 2. Comparing this curve with the superconducting ones, it can be noted that the M(H) signal at T = 15 K is not negligible, especially at high magnetic fields and high temperatures. Nevertheless, the width of M(H) in the superconducting state is much larger than the hysteresis of the magnetic curve at T = 15 K. However, before calculating the critical current density J_c_, the magnetic contribution was subtracted from the superconducting hysteresis loops, by using an analogous procedure to the one reported in Ref. [48], in order to be completely sure that it does not influence the calculation of J_c_.

At this point, the critical current density as a function of the magnetic field J_c_(H) was extracted at different temperatures by using the Bean critical state model [49,50]:(1)$$J_{c}=\frac{20\Delta M}{\left[c\left(1-\frac{c}{3b}\right)\right]}$$
where ΔM = M_dn_ − M_up_ is the difference between the magnetization measured for decreasing (M_dn_) and increasing (M_up_) applied fields, respectively. b (cm) and c (cm) are the length and width of the cross section of the crystal perpendicular to the applied field. The obtained J_c_(H) curves are reported in Figure 3. Observing the curves in the main panel and in the inset of Figure 3, it can be noted that a second magnetization peak phenomenon appears for T ≤ 7 K which is not visible at first sight when looking at the M(H) curves. In general, the J_c_(H) curves have a field decrease that prevents determining the irreversibility field H_irr_ (evaluated as J_c_ ≈ 100 A/cm^2^) even for the highest temperature shown in Figure 3 (11 K) and at 9 T.

In order to deeply study the J_c_(H) anomalous behavior reported in Figure 3, the J_c_ curves as a function of temperature J_c_(T) at different fields were extracted from the J_c_(H). In particular, by fitting the J_c_(T) behavior with several pinning models reported in the literature [51,52,53,54,55,56,57], it is possible to determine the pinning regime acting in the sample. Among the pinning models, the three equations that best fit our experimental data across the field range are the following ones:(2)$$\mathrm{Weak}\;\mathrm{pinning}:\;J_{c}^{\mathrm{weak}}\left(T\right)={\;J}_{c}^{\mathrm{weak}}\left(0\right){\;e}^{-T/T_{0}}$$
(3)$$\mathrm{Strong}\;\mathrm{pinning}:\;J_{c}^{\mathrm{str}}\left(T\right)={\;J}_{c}^{\mathrm{str}}\left(0\right){\;e}^{-3{\left(T/T^{*}\right)}^{2}}$$
(4)$$\mathrm{Weak}+\mathrm{strong}\;\mathrm{pinning}:\;J_{c}\left(T\right)={\;J}_{c}^{\mathrm{weak}}\left(0\right){\;e}^{-T/T_{0}}+J_{c}^{\mathrm{str}}\left(0\right)e^{-3{\left(T/T^{*}\right)}^{2}}$$
where $J_{c}^{\mathrm{weak}}\left(0\right)$ is the value of J_c_ at T = 0 K, and T_0_ is the characteristic pinning energy of weak (typically point-like) pinning defects [58,59,60]; $J_{c}^{\mathrm{str}}\left(0\right)$ characterizes the contribution to the J_c_ at T = 0 K and T* is the vortex pinning energy of strong pinning centers [59,61,62,63]. The absolute zero critical current approximation $J_{c}\left(0\right)$ is an important fitting parameter since its physical meaning arbitrarily marks the elimination of the thermal fluctuation effects due to the flux creep. Specifically, the fitting procedure has shown a weak pinning behavior for 0 T < $\mu_{0}$H ≤ 4 T due to point-like defects and a weak + strong pinning behavior for 5 T ≤ $\mu_{0}$H < 9 T due to the gradual activation of the twin boundaries present in this sample. In Figure 4, examples of the performed fit at different magnetic fields are reported while in Table 1 the fit parameters values are reported. In a very recent work on the same sample but in the H||c configuration [33], it has been demonstrated that the weak to strong pinning crossover in a sample presenting the second magnetization peak phenomenon triggers its onset. Here, it is worth underlining that the complete crossover to a strong pinning regime is not reached, indicating a delay in the complete vortex crossover into the strong defects. It is important to note that we are assuming the same triggering mechanism in the H||ab configuration since the results reported in Ref. [33] are independent of anisotropy.

From the fitting procedure, we can determine the J_c_(H) at zero temperature dividing its behavior in a weak and a weak + strong pinning region as reported in Figure 5. Moreover, the weak pinning region, highlighted in blue in Figure 5, can be fitted with the dependence expressed by the Kim model well [64,65,66] which is plausible for describing the field behavior of a superconductor in the presence of an homogenous distribution of point-like defects. From the fit reported in Figure 5 (red solid line), the zero field and temperature critical current density J_c_(0,0) can be extracted: J_c_(0,0) ≈ 6.42 × 10^5^ A/cm^2^.

In order to obtain more information about the type of defects present in the sample, the Dew-Hughes model [67] can be used. In particular, the normalized pinning force density F_p_ is calculated and plotted as a function of the reduced magnetic field (*h* = H/H_irr_):(5)$$F_{p}/F_{p}^{\max}={Ch}^{p}{\left(1-h\right)}^{q}\;$$
where C, *p* and *q* are fitting parameters that allow individuation of the pinning type of the material. Equation (5) takes into account a maximum in the F_p_ vs. *h* behavior. In particular, for *δl* pinning, the F_p_/F_p_^max^ maximum occurs at h_max_ = 0.33 with C = 27.8, *p* = 1 and *q* = 2 in the case of point pins, at h_max_ = 0.20 with C = 3.5, *p* = 0.5 and *q* = 2 in the case of surface pins, while no maximum occurs with C = 1, *p* = 0 and *q* = 2 in the case of volume pinning. For *δT_c_* pinning, the maximum is expected for higher *h* than *δl* pinning (see Ref. [67]). Therefore, to use Equation (5), it is necessary to know the irreversibility field but, as mentioned before, this is not possible for the J_c_(H) curves up to 11 K. For this reason, the J_c_(H) values at T = 12 K have been calculated (see inset of Figure 6). For T = 12 K, J_c_ is approximately equal to 100 A/cm^2^ at $\mu_{0}$H_irr_ = 2.6 T (see the blue arrow in the inset of Figure 6) and so it is possible to apply the Dew-Hughes method. The result is reported in the main panel of Figure 6 where F_p_/F_p_^max^ vs. *h* behavior is shown.

The fit of Equation (5) with the experimental data gives *h_max_* ≈ 0.2 with C = 3.33, *p* = 0.49 and *q* = 1.86, thus indicating that the surface pins dominate the pinning mechanism inside our samples at T = 12 K. Finally, since the sample has shown a second magnetization peak phenomenon, it is interesting to study the field dependence of the pinning force at different temperatures and to calculate the pinning force values, comparing them with the literature. The results are reported in Figure 7. It can be noted that the F_p_ values decrease with increasing temperature following the J_c_ behavior. On the other hand, it is worth underlining that for T < 9 K, the pinning force curves have a monotonic increasing trend with increasing magnetic field. This feature could be exploited since these temperatures are typically considered for power applications of superconductivity. Moreover, comparing the F_p_ values with those reported in the literature [16,38,68,69], it can be noted that they are similar to the values reported for other Fe(Se, Te) single crystals (10^7^ ÷ 10^9^ N/m^3^) but they are much higher if compared with bulk and polycrystalline Fe(Se, Te) samples (10^5^ ÷ 10^7^ N/m^3^). It is worth underlining that our F_p_ values have been compared with the H||c field configuration reported in the literature. This is not a problem since it is well known that Fe(Se, Te) is a weakly anisotropic material [6,70,71]. It is also an interesting observation that IBS systems tend to show stronger pinning abilities in samples with high crystalline morphology, such as thin films [11,72,73,74], due to the effective naturally formed disorder. This will have a positive effect on the power stability and performance of various nano/micro-superconducting devices which similarly are affected by the vortex motion. These high pinning force values together with the presence of the second magnetization peak phenomenon indicate that the material can be promising for high-current and high-power applications.

## 4. Conclusions

We have studied an Fe(Se, Te) twinned single crystal fabricated by the Bridgman technique by analyzing the dc magnetic measurements as a function of temperature and magnetic field. In particular, the magnetic field was applied in a parallel field configuration H||ab. By using M(T) measurements, we have obtained T_c_ = 14.5 K and noted the presence of a magnetic background, probably due to magnetic impurities present in the sample. A magnetic background was also observed in the superconducting hysteresis loops M(H) performed at different temperatures which showed a tilt in their behaviors. After subtracting the magnetic signal, the critical current density J_c_ at different temperatures was extracted from the M(H) curves, showing the presence of a second magnetization peak phenomenon which allowed the sample to sustain high J_c_ values even at high magnetic fields. Extracting the J_c_(T) curves from the J_c_(H) ones, we analyzed them in terms of weak and strong pinning regimes acting in the sample. Based on the Kim model analysis, it was found that in the parallel field geometry, the Fe(Se, Te) crystal (as for H||c in our previous studies) similarly undergoes a pinning crossover from a weak pinning regime, ascribed to planar point-like defects, to a weak + strong pinning regime due to the gradual activation of the twin boundaries. However, in this case the SMP features are much broader and the consequent non-monotonous peak change of Jc is observed only in certain temperature (closer to Tc) and high field ranges. After that, using Dew-Hughes analysis, we identified that the dominating pinning mechanism from surface (planar) defects affects the vortex system at 12 K. Finally, we calculated the pinning force density F_p_ values, noting that they have an interesting monotonous increasing trend as a function of magnetic field at temperatures exploitable in practical situations. The F_p_ values of the sample were compared with the ones reported in the literature, noting that they are much higher with respect to the polycrystalline and bulk sample values, confirming the suitability of the sample in its use for high-power applications.

## Figures and Tables

**Figure 1 materials-14-05214-f001:**
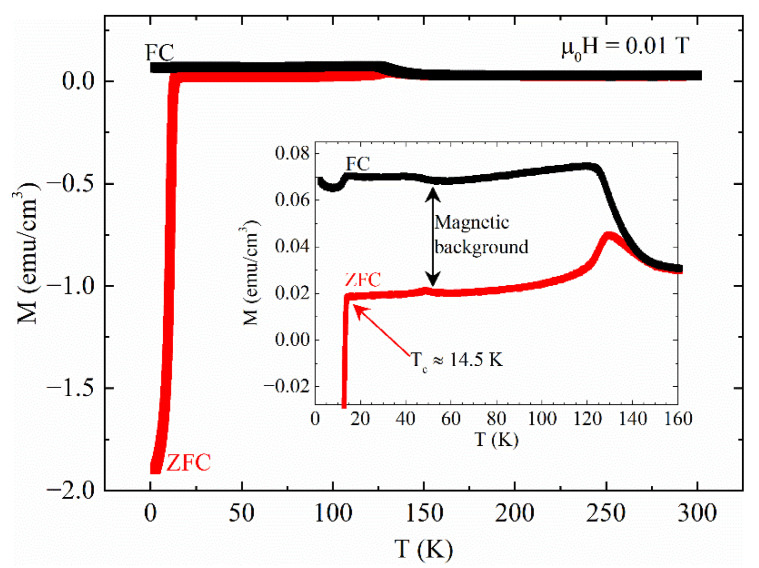
Magnetization as a function of temperature M(T) measured in ZFC-FC conditions with an applied magnetic field $\mu_{0}$H = 0.01 T. Inset: a magnification of the temperature region between 0 K and 160 K shows the presence of a magnetic background. The red arrow indicates the T_c_ of the sample.

**Figure 2 materials-14-05214-f002:**
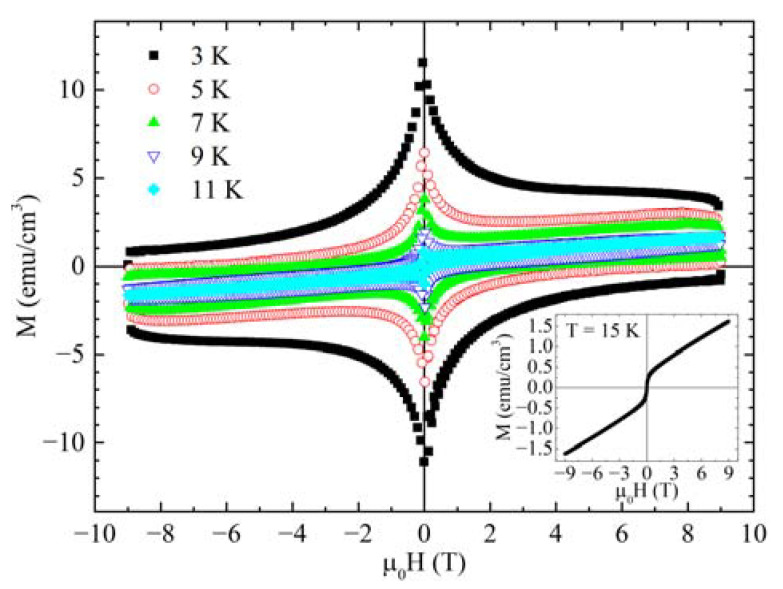
Superconducting hysteresis loops at different temperatures. Inset: magnetic hysteresis loop at T = 15 K.

**Figure 3 materials-14-05214-f003:**
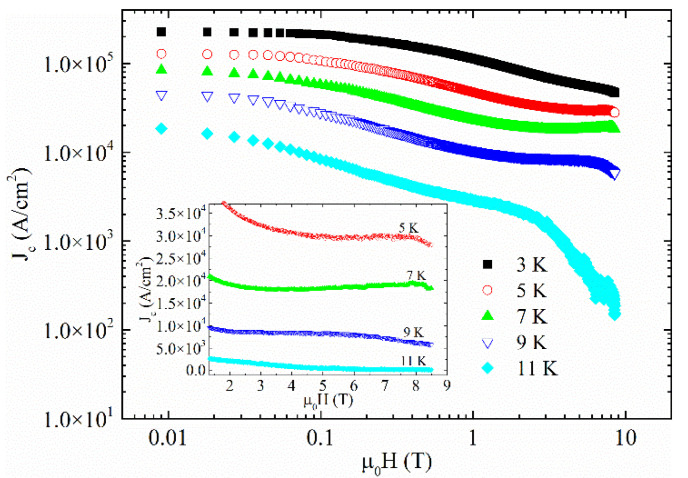
Critical current densities as a function of field at different temperatures. Inset: a magnification of the curves near the region of the second magnetization peak phenomenon is shown.

**Figure 4 materials-14-05214-f004:**
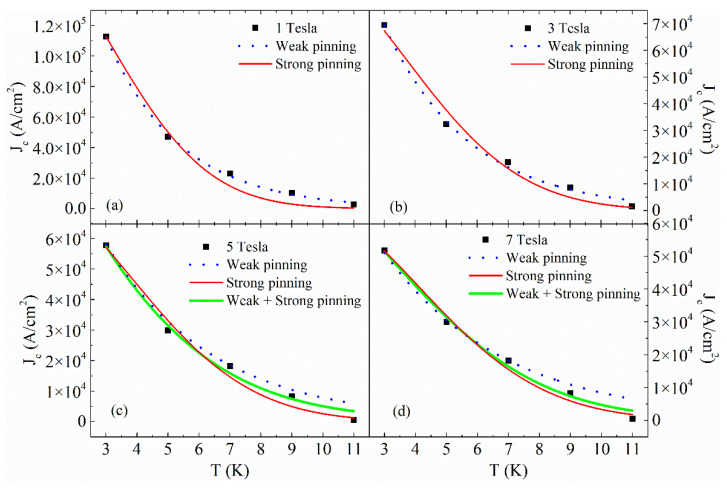
Temperature dependence of J_c_ at $\mu_{0}$H = 1 T (**a**) and $\mu_{0}$H = 3 T (**b**) fitted with weak pinning equation (blue dotted line) and strong pinning equation (red solid line); temperature dependence of J_c_ at $\mu_{0}$H = 5 T (**c**) and $\mu_{0}$H = 7 T (**d**) fitted with the weak pinning equation (blue dotted line), the strong pinning equation (red solid line) and the combination of weak and strong pinning equations (solid green line).

**Figure 5 materials-14-05214-f005:**
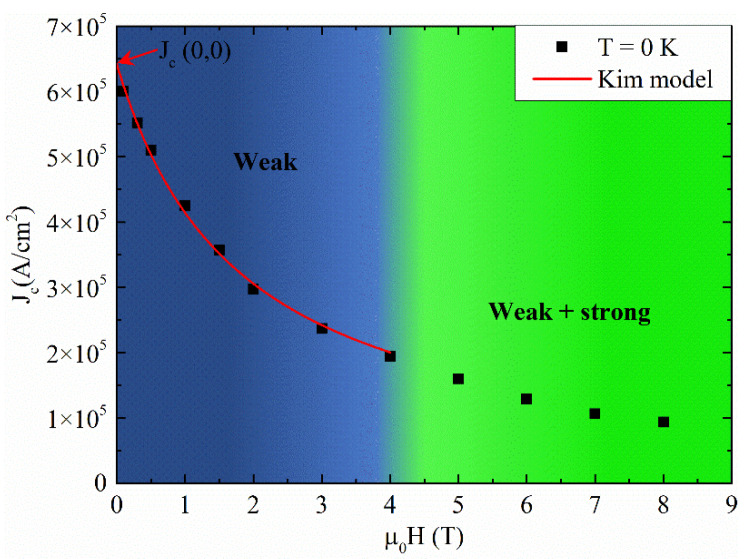
Magnetic field dependence of J_c_ at T = 0 K. The red solid line is the fit of the J_c_(H) at T = 0 K with the Kim model.

**Figure 6 materials-14-05214-f006:**
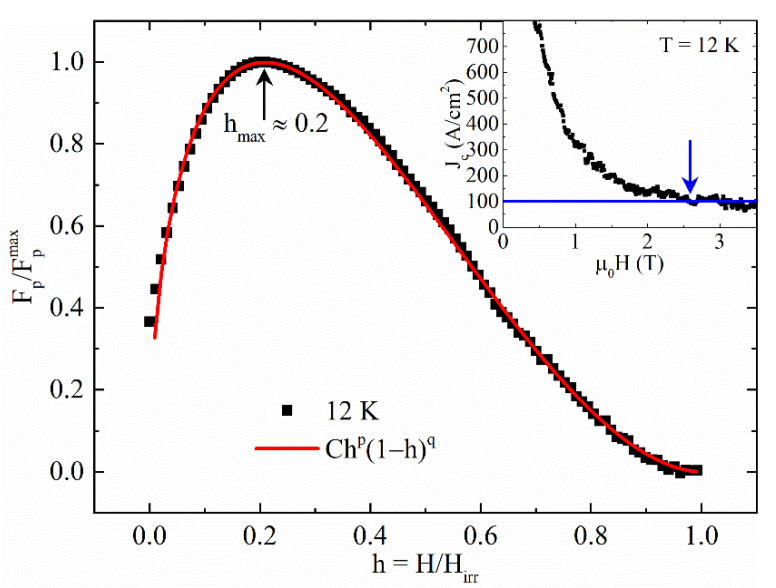
Normalized pinning force density ($F_{p}^{\max}\approx 3.9\times 10^{6}N/m^{3})$ as a function of the reduced magnetic field H/H_irr_ at T = 12 K fitted with Equation (5) (red solid line). Inset: determination of H_irr_ at T = 12 K ($\mu_{0}$H_irr_ = 2.6 T).

**Figure 7 materials-14-05214-f007:**
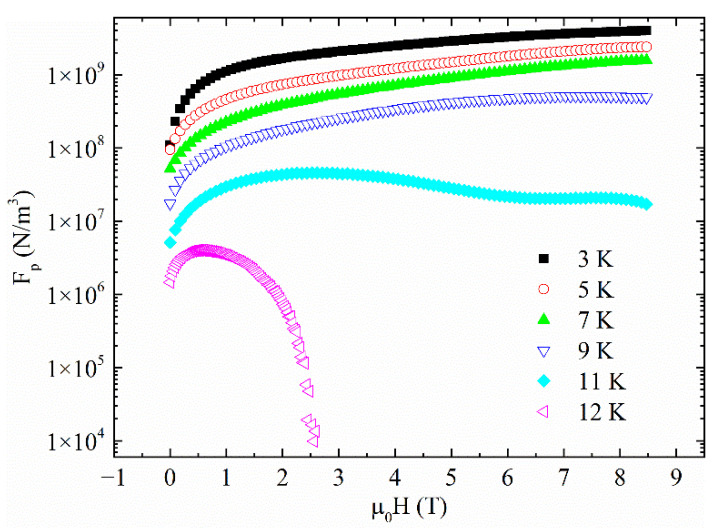
Pinning force density as a function of field at different temperatures.

**Table 1 materials-14-05214-t001:** Parameters values of the fit procedure performed in Figure 4.

Fit Parameter	$\mu_{0}H=1\;T$	$\mu_{0}H=3\;T$	$\mu_{0}H=5\;T$	$\mu_{0}H=7\;T$
$J_{c}^{\mathrm{weak}}\left(0\right)$(A/cm^2^)	390,640	205,490	137,140	110,070
T_0_ (K)	2.41	2.76	3.5	3.9
$J_{c}^{\mathrm{str}}\left(0\right)$(A/cm^2^)	177,780	93,400	77,455	67,570
T* (K)	7.70	9.09	9.40	10.00

## Data Availability

The data sets that support the findings in this study are available from the corresponding author upon reasonable request.

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
