# Peer review of "High Pinning Force Values of a Fe(Se, Te) Single Crystal Presenting a Second Magnetization Peak Phenomenon"

_materials, 2021, doi:10.3390/ma14185214_

Round 1
Reviewer 1 Report
I read the manuscript entitled “High pinning force values of a Fe(Se,Te) single crystal presenting a second magnetization peak phenomenon” by A. Galluzzi et al, submitted for a publication in materials. The authors report magnetization curves of one of the iron-based superconductors and analyzed the data using different models to extract parameters relevant to the applications of superconductivity.
Iron-based superconductors are promising for the applications of superconductivity due to their reasonably high critical current density and FeSe has been object of several studies for its structural simplicity. Here, the authors have taken FeSe0.5Te0.5 (actual composition Fe0.96Te0.59Se0.45) single crystal as model and studied the magnetization curves as a function of temperature. The work is very similar to the one reported by the same authors in their earlier publications [31, 32] (Scientific Reports 11, 7247 (2021), A Galluzzi et al 2018 Supercond. Sci. Technol. 31 015014) containing some extension of the analysis. Nevertheless, the manuscript still contains minimal ingredients that could be useful for very specialized readers. I recommend the publication of the manuscript with minor revision:
1) I think the authors should explicitly mention what is new in this manuscript with respect to the two earlier papers [31, 32] reporting similar data on the same sample.
2) Fig. 5 shows the crossover from weak to weak+strong pinning crossover. As plotted, it looks misleading, i.e. is there a clear boundary that separates the two regimes. What about the strong pinning regime?
3) The authors have mentioned 0 K values, however, it is advised to state clearly what they mean by 0 K values.
4) For a better readability the authors are advised to include quantitative values of the pinning forces reported in the literature for polycrystalline and single crystal samples for comparison and a ready reference for the readers.
5) Typos are popping up in the text; e.g., the phrase “The obtained Jc(H) curves are reported in Figure 3.” is repeated in the text.
Author Response
Thank you for the critical comments and valuable suggestions.
Please see the attachment.

Reviewer 2 Report
Manuscript Number: materials-1335363
Title High pinning force values of a Fe(Se,Te) single crystal presenting a second magnetization peak phenomenon
Authors: Armando Galluzzi, Krastyo Buchkov , Vihren Tomov , Elena Nazarova , Antonio Leo , Gaia Grimaldi , Massimiliano Polichetti
This manuscript presents careful experimental approach and detailed results on the study of high pinning force values of a Fe(Se,Te) single crystal presenting a second magnetization peak phenomenon. This will serve as a valuable reference for future work on similar systems.
The manuscript contains enough significant new physics and scientifically sounds.
The paper is well organized and clearly written.
The subject matter is appropriate for Materials.
The referee is convinced that the scientific content of the manuscript is suitable to be published.
Author Response
We would like to thank the Reviewer for his/her positive comments about our Manuscript.
Reviewer 3 Report
I have read with interest the manuscript.
My feeling is that the manuscript presents some interesting results, and it should be published after some minor revisions.
In the following, I discuss the points that in my opinion should be improved. Whenever possible, I refer to the line number.
l.3, l.21-22, l.58-59, l.132 The authors call the nonmonotonous field dependence of Jc, as extracted from M(H) loops, as a ”second peak magnetization phenomenon”. I do not think that it is correct. No second peak is present in the magnetization. Although I am aware that the phenomenon reported by the authors has been called with this name (”second magnetization peak”), it is not a sufficient motivation to keep the misuse. I suggest that this is called as it is, ”peak in Jc” or ”nonmonotonous Jc”. I then suggest to change the title and the text accordingly.
l.23 The word ”individuated” does not sound well in English.
l.24 Same for ”birth”
l.39 ”Although their low Tc” should be changed to, e.g., ”Despite their low Tc”
l.41-42 In fact, it is not just the Jc, H_irr and H_c2 high values that make Fe-based SC interesting for applications. The good connectivity (grain boundaries properties) are possibly as important. The authors might want to mention this aspect with some relevant citations.
l.47 Tc=14.5 K is perhaps Tc in the sample here studied, but it is not the representative Tc of the whole Fe(Se,Te) family (e.g., with different Se:Te ratio), as one is led to believe from the text. Please rewrite the sentence.
l.51 Please remove ”the” before ”Fe(Se,Te)”
l.65 The sentence ”we have investigated the pinning landscape” is not completely clear to me. It looks like a spatial profile of the pinning potential has been depicted, which is clearly not in the manuscript. Maybe the authors meant that they investigated the different kind of pinning centers (δl, δTc, ….). Please clarify.
l.86-87, and throughout the manuscript: in principle, H is not measured in tesla, although this misuse is widespread. I do not suggest to change everything to A/m, since it is against the common use, but I think that a correction is required. The authors might choose to use B (but they should specify that it is an ”applied” flux density, not the local B: it can be rather different in the Bean penetration regime, and then they might choose a symbol like B_a), or use μ_0H where relevant (basically in the x axis of some figures, and in a few places in the text), so they can still refer to ”M vs. the magnetic field” or ”M(H)” (B is not the magnetic field, but the filed induction or flux density). In the Materisl and Methods section they can define the field and the symbol, and make the change in the manuscript (including legends, figures…).
l.99-100 The magnetic background is likely to be insensitive to the field orientation (there will be geometrical effect, however). Since the authors previously published results on a very similar (or maybe the same) sample (their ref.31), this information should be reported.
l.113 What the ”width of the superconducting curves” is? The width of which quantity vs. which? Better say ”The width of M(H) in the superconducting state is much larger…”.
l.114 ”much larger” sounds better than ”much higher” (the sentence is about a width).
l.128-129 the lengths b and c: are they the sizes of the crystal, that is 3 and 0.2 mm? if yes, they do not ”characterize the cross section”, but they ”are the length and width of the cross section of the crystal”. If not, it should be discussed why not and what are those lengths, if there is any correction with respect to the geometry.
l.132 The nonmonotonous Jc appears only at low T, it should be mentioned.
l.133 m(H) should be capitalised as M(H)
l.133-134 the sentence ”smooth field dependence decrease proven by the impossibility to determine” is not clear. It should be rephrased, e.g. as ”smooth field decrease that prevents from determining”
l.147-150 Please add specific references for Eqs. 2-4, not bundled together.
l.153-157 and Fig.4: Please show the comparison of the weak vs strong fits in Fig.4 a and b, and weak vs. weak+strong fits in Fig 4c and d, to appreciate how one fit is better than another. A quality index would be welcome (chi squared, other)
l.157-159 the experimental result mentioned was obtained with H//c. It should be mentioned, and possibly discussed whether the authors believe that this is a general feature irrespective of anisotropy.
l.162-165, Fig.4 Since fits are performed, the fit parameters should be reported as a plot or a table: Jc_str(0), Jc_weak(0), T*, T_0. Since it is suggested that a second (strong) pinning mechanism arises in certain circumstances, the significance of the fits, and then the conclusions drawn from them, will be reinforced from a smooth temperature dependence of the fit parameters (in particular, those related to the weak contribution).
l.180 C is not a generic constant, but an explicit expression of p and q. It should be emphasized that it is not an additional fitting parameter, and its value should be reported (maybe in the caption of Fig.6)
l. 191 Please do not use the Oe unit. Apart from being a non-SI unit, H has never been expressed in Oe elsewhere in the manuscript. See a previous comment on the magnetic units.
l.195-196 please report the values for Fp_max and H_irr in the caption of Fig.6
l.197-198 This conclusion only holds at T=12 K, please specify
l.201-203 This way to derive Fp is standard. Moreover, it is placed after Eq-6, where Fp is obtained with this method. This description it should not be placed here but in the methods section.
l.208-209 Is the comparison with literature made with data in the same orientation, H//ab? If not, the comparison might be less significant due to the anisotropy of Fe(Se,Te). Please specify, and in case please discuss.
l.214-215 Fig.7. Please plot also the data at 12 K in this same figure (replot of the data in Fig.6, not normalized, as a function of H). Moreover, although at first sight the curves do not scale, the authors should try to scale the low-field parts of the curves using a h_irr and F_p_max as parameters using as a guide the 12K curve. It is likely that they will not able to scale the entire curves, but) they could however find that the low field part can be scaled, and from the departure from the scaling they can find the crossover field H_onset(T) where a second pinning mechanism sets in.
l.231-232 This conclusion refers to the 12K data only. It should be mentioned. Moreover, the authors should discuss the possible consequences on the performances in thin films, that are interesting for some applications.
Author Response

(The authors gave the same response as above.)
